# Coiled-Coil N21 of Hpa1 in *Xanthomonas oryzae* pv. *oryzae* Promotes Plant Growth, Disease Resistance and Drought Tolerance in Non-Hosts via Eliciting HR and Regulation of Multiple Defense Response Genes

**DOI:** 10.3390/ijms22010203

**Published:** 2020-12-28

**Authors:** Zhao-Lin Ji, Mei-Hui Yu, Ya-Yan Ding, Jian Li, Feng Zhu, Jun-Xian He, Li-Na Yang

**Affiliations:** 1College of Horticulture and Plant Protection, Yangzhou University, Yangzhou 225009, China; zhlji@yzu.edu.cn (Z.-L.J.); ydnwkxk@163.com (Y.-Y.D.); yz_lijian@163.com (J.L.); zhufeng@yzu.edu.cn (F.Z.); 2School of Life Sciences and State Key Laboratory of Agrobiotechnology, The Chinese University of Hong Kong, Shatin, New Territories, Hong Kong, China; cketnn7@hotmail.com

**Keywords:** N21_Hpa1_, HR, disease resistance, growth promotion, drought tolerance

## Abstract

Acting as a typical harpin protein, Hpa1 of *Xanthomonas oryzae* pv. *oryzae* is one of the pathogenic factors in hosts and can elicit hypersensitive responses (HR) in non-hosts. To further explain the underlying mechanisms of its induced resistance, we studied the function of the most stable and shortest three heptads in the N-terminal coiled-coil domain of Hpa1, named N21_Hpa1_. Proteins isolated from *N21*-transgenic tobacco elicited HR in Xanthi tobacco, which was consistent with the results using N21 and full-length Hpa1 proteins expressed in *Escherichia coli*. N21-expressing tobacco plants showed enhanced resistance to tobacco mosaic virus (TMV) and *Pectobacterium carotovora* subsp. *carotovora* (*Pcc*). Spraying of a synthesized N21 peptide solution delayed the disease symptoms caused by *Botrytis cinerea* and *Monilinia fructicola* and promoted the growth and drought tolerance of plants. Further analysis indicated that N21 upregulated the expression of multiple plant defense-related genes, such as genes mediated by salicylic acid (SA), jasmonic acid (JA) and ethylene (ET) signaling, and genes related to reactive oxygen species (ROS) biosynthesis. Further, the bioavailability of N21 peptide was better than that of full-length Hpa1_Xoo_. Our studies support the broad application prospects of N21 peptide as a promising succedaneum to biopesticide Messenger or Illite or other biological pharmaceutical products, and provide a basis for further development of biopesticides using proteins with similar structures.

## 1. Introduction

Harpin proteins are virulence factors in susceptible plants, translocators for effectors, elicitors of hypersensitive response (HR), inducers of defense responses and enhancers of plant growth in non-host plants [1,2,3,4,5,6,7,8,9]. Hpa1 is one of the harpin proteins that is secreted via type III secretion systems into the extracellular spaces of plant cells [1,4,10,11].

Hpa1 of the Gram-negative bacteria *Xanthomonas oryzae* pv. *oryzae* which causes the bacterial blight of rice consists of 139 amino acids with two predicted α-helices at the N-terminal (36–53 aa) and C-terminal (87–103 aa), respectively, and has a high glycine content, particularly in the middle and C-terminal [4,9,12]. The functions of the different domains of some harpin proteins have been studied. The N-terminal of Hpa1_Xoo_ has bioactivities, including inducing resistance and promoting growth in tobacco, and leaf photosynthesis in *Arabidopsis* [12,13,14,15,16]. N-terminals that lacked 36 amino acids or lacking one or two of the α-helices of Hpa1_Xoo_ exhibit reduced virulence, and both α-helices play a critical role in the translocation of transcription activator-like (TAL) effectors [9]. The α-helix of the N-terminal of HpaG of *X. axonopodis* pv. *glycines* is essential for eliciting an HR in tobacco [17]. The C-terminal 216 amino acids of HrpZ of *Pseudomonas syringae* pv. *syringae*, 200–300 residues of HrpZ of *P. syringae* pv. *phaseolicola* and C-terminal 21 residues of Hpa1_Xoo_ could also elicit a strong HR [12,18,19]. However, the coiled-coil (CC) regions in the N-terminal and C-terminal of Hpa1_Xoo_ have the opposite functions: CC-formation in the N-terminal could induce HR but CC-formation in the C-terminal could not, since only a polymerized CC structure could cause HR [12].

CC domains consist of two or more α-helices that interact with each other to form supercoiled bundles via “knobs-into-holes” (KIH) interactions [20,21,22]. Typical CC domains are contiguous heptad repeats (seven amino acids repeats), often defined as *abcdefg*, in which the *a* and *d* hydrophobic residues provide driving forces for intertwining, and influence the oligomerization of the helices [23,24,25]. The *e* and *g* sites balance the opposing forces of the α-helices and maintain the stability and oligomerization of the structure [26,27]. Two heptads could form high degree of α-helices and 100% dimers in the corresponding buffers. However, the oligomerization state may change from dimers to trimers with increasing ionic strength [28,29]. Therefore, three heptads are the shortest sequences to stabilize CC folding. CC domains are important assembly units in a variety of structures and regulatory proteins in eukaryotes and prokaryotes, such as transcription regulators, membrane sensors and skeletal proteins [26,30,31]. Furthermore, our previous work has indicated that CC domains play essential roles in structure of *Xanthomonas* Hpa1 protein which could induce HR in non-host. Three heptads of the α-helix in the N-terminal of Hpa1_Xoo_, named N21, form a coiled-coil domain that was a mixture of dimers and monomers and induced strong HR in tobacco. Trimers of 21 aa in the C-terminal of Hpa1_Xoo_ induce relatively minor HR in tobacco, but it has a lower probability to form CC region [12]. Wang and his colleagues have showed that mutation of 12 highly hydrophilic amino acids in the N-terminal of Hpa1_Xoo_ abolished the ability of Hpa1_Xoo_ to elicit HR in tobacco and that these conserved amino acids played critical roles in protein aggregation [13,32]. All these evidences suggest that N-terminal of Hpa1_Xoo_ and formation of CC domain are important for protein function. Therefore, we focused our research on coiled-coil 21 amino acids at the N-terminal of Hpa1_Xoo_.

Harpin proteins, as Messenger from *Erwinia amylovora*, that was popularized by US environmental protection agency, and Illite from *Xanthomonas oryzae* pv. *oryzicola* (*Xoc*), that was studied in a plant pathology laboratory in Nanjing Agriculture University, were developed as biopesticides, because of their promoting effects on plant growth, crop yield, plant disease resistance, non-toxicity, no residue and environmental friendliness. However, poor bioavailability was the major constraining factor for the use of harpins as biopesticides because few harpin molecules interacted with receptors due to the particular structure of plant leaves. Podile and co-workers wrapped harpin_Pss_ in chitosan nanoparticles to improve its bioavailability and disease resistance in tomato [33].

N21_Hpa1_ induced HR in tobacco, as effective as the full-length Hpa1_Xoo_, thus we wanted to learn more about the function of N21 to see if it could be used as a succedaneum to Hpa1_Xoo_. Results of the present work suggested that protein of transgenic-*N21* tobacco plants retained sufficient activity to elicit HR. *N21*-expressing tobacco plants enhanced the resistance to tobacco mosaic virus (TMV) and *Pectobacterium carotovora* subsp. *carotovora* (*Pcc*). The application of an N21 peptide solution to some plants induced several beneficial effects, including improvement of host resistance to *Botrytis cinerea* and *Monilinia fructicola*, promotion of plant growth, enhancement of drought tolerance that is even better than that of the full-length protein. In addition, N21-induced multiple defense responses in transgenic-*N21* tobacco and had a better bioavailability.

## 2. Results

### 2.1. Generation of Transgenic-N21 Tobacco and Trans-N21 Protein Activity Assay

The expression vector was constructed in the digested skeleton of pBI121 using the cDNA sequences of *N21* (Figure 1A). The constructed vector was first transformed into *Agrobacterium* strain EHA105 and then co-cultured with tobacco plants by the leaf disc method (Appendix A). The progeny of transgenic tobacco was sub-cultured to the T3 generation to screen for recombinant protein expressing plants via kanamycin resistance and further verified by using PCR (Appendix A) and semi-quantitative RT-PCR (Appendix A). The selected T3 plants were further screened for homozygotes. Phenotypic observation on the selected *N21*- and *Hpa1*-expressing tobacco plants showed that the transgenic plants grew much bigger than the control plants transformed with the empty vector (EV or *pBI121*-expressing) (Figure 1B). Root length of the trans-*N21* tobacco was ~1.3-fold higher than that of EV-ones and ~0.8-fold lower than that of trans-*Hpa1* plants (Figure 1E). The plant height of trans-*N21* tobacco was ~0.8-fold that of the full-length transgenic tobacco, and ~1.5-fold of the EV tobacco (Figure 1F). The fresh weight of trans-*N21* was ~2.5-fold of that of EV and ~0.7-fold of trans-*Hpa1* tobacco (Figure 1G). Total protein extracts from the *N21*-expressing tobacco were injected back into WT Xanthi tobacco plants. Equivalent amount of N21 and Hpa1 proteins expressed directly in *Escherichia coli* (*E. coli*) were used as the positive control. Results showed that the N21 protein expressed in trans-*N21* tobacco retained sufficient activity to induce HR in Xanthi tobacco. However, the activity was lower than the Hpa1 and N21 proteins expressed by *E. coli* (Figure 1C). The lower activity might be due to the lower concentration of N21 in trans-*N21* tobacco. The measured diameters of HR activity zones were consistent with the above results (Figure 1D).

### 2.2. N21-Expressing Tobacco Showed Enhanced Resistance to TMV and Pcc

The resistance of tobacco to TMV can be rapidly induced by harpin with a stable defense response. Therefore, this response is usually used as a basic index of plant disease resistance induced by harpin proteins [12,34]. We investigated whether N21 had a similar function. TMV was inoculated on different transgenic tobacco leaves and the disease symptoms were observed at 36 and 72 h post inoculation (hpi). The results showed that the number of lesions on N21- and Hpa1-expressing tobacco were significantly reduced compared to the EV tobacco at 36 hpi, and the disease resistance reflected by the lesion reduction rate was 70.5% (N21) and 72.3% (Hpa1), respectively. At 72 hpi, the trans-*N21* and trans-*Hpa1* plants had 75.1% and 80.9% of the lesion reduction rate, respectively, compared to the EV plants (Figure 2A and Table 1). The results of virus content assays in diseased tobacco leaves showed that the content of TMV was the least in trans-*N21* tobacco at 36 hpi, with no significant difference between trans-*N21* and trans-*Hpa1* plants at 72 hpi, both showing lower virus contents than that in EV tobacco (Figure 2B). These results indicate that N21-expressing tobacco had also improved resistance to TMV and that the resistance level was higher than that of transgenic tobacco expressing the full-length *Hpa1* at the early stage of infection.

*Pectobacterium carotovora* subsp. *carotovora* (*Pcc*) is a pathogenic-bacteria that spreads through plant leaf veins to cause disease. The leaves of different transgenic tobacco plants were impregnated with a suspension of *Pcc* and compared with plants immersed in water. At 12 hpi, we did not see any differences between the *Pcc*- and water-treated plants either in the trans-*N21* or trans-*Hpa1* plants. However, bacteria were detected by quantitative real-time PCR (qRT-PCR) in *N21*- and *Hpa1*-transgenic tobacco, and found that the content of *Pcc* in trans-*N21* tobacco was the lowest one as compared with that in EV and trans-*Hpa1* tobacco and the differences were significant. Meanwhile, the main vein and some vein branches of EV plants became bigger differences (Figure 3A,B, white arrows). The differences at 16 hpi were larger than at earlier times, since most leaf veins of EV plants exhibited obvious soft rot, but trans-*N21* and trans-*Hpa1* plants only had a small portion of watery veins (Figure 3A). The results of bacteria content assay showed a similar trend (Figure 3B). These results suggest that both trans-*N21* or trans-*Hpa1* plants inhibited the infection rate of *Pcc* via leaf veins of tobacco and thus enhanced the resistance to *Pcc*.

### 2.3. Treatment with a N21 Peptide Solution Increased Host Resistance to M. fructicola and B. cinerea

Based on the increased resistance of the N21-expressing tobacco to TMV and *Pcc*, we synthesized and purified the N21 peptide, via *F-moc* synthesis, in order to further study the function of N21. We measured the resistance of peach plants to *Monilinia fructicola*, a pathogen that causes peach brown rot. The peach plants were pre-treated with different solutions for 24 h, including water, a N21 peptide solution (N21-PS) at the concentration of 40 μg/mL, and procymidone (~0.5 mg/mL) (a prevention agent of peach brown rot. Following each pre-treatment, hyphal blocks of *M*. *fructicola* were inoculated on the wounded surface of the peach and the incidences were observed at different time points. The results showed that peaches treated with sterile water showed the largest rotting area and mildew growth at 24 hpi. Plants treated with procymidone exhibited minor decay, with 38.2% inhibition of the pathogen, and the plants treated with the N21-PS exhibited the smallest rotting area at 24 hpi, with an inhibition rate of 59.6%. With the extension of inoculation time (72 hpi), the entire fruit of the water-treated peach was invaded by *M*. *fructicola* with abundant mycelia. However, for plants treated with N21-PS, ~36.7% of the peach area was rotten with distinct hyphae. In plants treated with procymidone, about ~52.7% of the peach area were infected by mycelia of *M. fructicola* (Figure 4A,B). These results indicate that the N21 peptide was more effective in inhibiting the brown rot of peach compared to procymidone.

In another experiment, the N21-PS (40 μg/mL), carbendazim (800 times dilution, ~0.88 mg/mL) and sterile water was respectively used to spray the surface of strawberries and tomatoes. After standing the plants for 24 h in a lighted incubator at 25 °C, conidial suspensions (5 × 10^5^ spores/mL) or blocks of *B. cinereal* (the pathogen inducing grey mould) were inoculated on the wounded surface of strawberry and tomato plants and cultivated in a chamber at 25 °C with 50% humidity. The results revealed that the strawberries and tomatoes treated with the N21-PS and carbendazim showed delayed occurrence time of grey mould compared to the water-treated control. The inhibition rate of *B. cinerea* by the N21-PS was 31.3% in strawberry on the fourth day after inoculation, which was smaller than the inhibition effect of carbendazim (~54.3%). However, opposite effects were observed in tomato, where higher inhibition rate was seen in treatments with N21-PS than with carbendazim (Table 2). All these results demonstrate that N21 peptide can delay the disease occurrence time and increase the resistance to grey mould.

### 2.4. N21 Peptide Promoted the Growth of Plants

Previous studies demonstrated that harpins have growth-promoting effects and can increase the drought resistance in several plant species [35,36,37]. Therefore, we investigated whether N21 peptide had the similar effects. To test its growth-promoting effect, tomato plants grown in pots were sprayed with sterile water or N21-PS (40 μg/mL) every ten days, and then allowed to grow for 45 days before observation. The results showed that the tomato plants sprayed with the N21-PS reached a plant height of ~49.0 cm, fresh weight of ~20.1 g and root length of ~11.7 cm, which represent 11.6%, 19.1% and 7.09% of increase, respectively, compared to the water-sprayed plants (Figure 5 and Table 3).

Then, we tested the growth-promoting effect of N21-PS in seedling of several plants. Sterilized seeds of tomato, pepper, cucumber, melon and wheat were dipped in the N21-PS (40 μg/mL) or sterile water and placed in an incubator at 25 °C with a 13 h light /11 h dark cycle for 5 days, and the root length was recorded. The root length of plants treated with the N21-PS were 12% (tomato), 24.8% (pepper), 51.4% (cucumber), 11.7% (melon) and 4.9% (wheat) higher than the water-treated plants, respectively (Table 4). The germinated seedlings of different plant species were planted in pots and sprayed with sterile water or N21-PS every two days and the plant height was measured on the tenth day. The results showed that N21-PS effectively promoted the growth of the plants treated, especially pepper, whose plant height was 29.4% higher than the water-treated plants. N21-induced increase of plant height was also seen in cucumber (20.9%), wheat (12.4%), tomato (12.3%) and melon (7.0%) (Table 4). Taken together, these results provided evidence that N21 peptide could also promote plant growth.

### 2.5. N21 Peptide Improved the Drought Tolerance of Tobacco

Polyethylene glycol 6000 (PEG6000) is usually used to test plants’ drought responses [36,38,39]. We used 10% PEG6000 to simulate drought stress in tobacco. 60-day-old tobacco seedlings were irrigated with 10% PEG6000 for 48 h and the leaves were sprayed with water, N21-PS (80 μg/mL) or the same amount of Hpa1 protein expressed from *E. coli* every day. The results showed that the plants treated with the N21-PS only exhibited mild wilting, and the leaves returned to normal after re-watering. However, tobacco plants sprayed with sterile water showed irrecoverable leaf wilting and signs of yellowing. Plants treated with the same amount of Hpa1 protein showed clear wilting symptoms, but milder than those treated with sterile water (Figure 6A). Consistently, the relative water content (Figure 6B), germination rate (Appendix A) and survival rate (Appendix A) of the plants, showed similar change patterns. A significantly higher expression of the drought-stress genes (*NtERD10B* and *NtLEA5*), genes for ROS detoxification (*NtSOD*, *NtAPX* and *NtCAT*) and signaling components (*NtPLC3* and *NtCMK1*) was observed in the plants treated with the N21-PS and Hpa1 proteins, relative to the water-treated plants, with highest expression in the N21-PS treatment (Figure 6C–I). All these results indicated that the N21 peptide could improve the drought tolerance of tobacco plants even better than the full-length Hpa1 protein.

### 2.6. Coiled-Coil N21 Upregulated the Expression of Multiple Defence Response Genes in Tobacco

To further examine the molecular mechanisms of the induced resistance of coiled-coil N21, we measured the transcription of 11 defense-related genes in N21-expressing, Hpa1-expressing and EV tobacco plants, including four salicylic acid (SA)-related genes (*PR-1a*, *PR-1b*, *PR2*, *NPR1*), two jasmonic acid (JA) synthesis genes (*LOX1*, *AOC4*), three ethylene (ET) synthesis and signaling genes (*ACS1*, *ACS2*, *EIN2*), and two reactive oxygen species (ROS) synthesis-related genes (*RBOHA*, *RBOHB*). The results showed that nearly all the tested genes had higher expression in trans-*Hpa1* tobacco compared to the trans-*N21* and EV tobacco plants. The *PR-1a*, *PR-1b*, *PR2*, *NPR1*, *LOX1*, *AOC4*, *ACS2*, *RBOHA* and *RBOHB* genes were upregulated in N21-expressing and Hpa1-expressing tobacco plants as compared with the EV tobacco plants (Figure 7A–D). However, the expression of some genes was significantly different between the trans-*N21* and trans-*Hpa1* plants. Among these, the expression of *PR-1a* in N21-expressing plants were ~30-fold higher than in Hpa1-expressing tobacco plants (Figure 7A). The expression of *PR-1b*, *PR2*, *LOX1*, and *RBOHA* was dramatically higher in trans-*Hpa1* tobacco than in trans-*N21* tobacco (Figure 7A,B,D). These results indicated that the coiled-coil N21 could induce an immune response in transgenic tobacco and play a similar role as Hpa1_Xoo_ in some of the defense responses, however, it also plays some different roles in defense as reflected by its different effects on gene expression.

### 2.7. N21 Peptide Has Better Bioavailability than Hpa1_Xoo_

The above results suggest that the N21 peptide has similar effects than the full-length Hpa1, including promoting plant growth, inducing HR in non-host, increasing disease and drought tolerance. The N21 effect in inducing drought tolerance was even better than the full-length Hpa1. Interestingly, the trans-*N21* tobacco also had lower -virus or bacterial content than that in trans-*Hpa1* tobacco in the early stage of infection, and showed similar resistance to trans-*Hpa1* in the late stage of infection (Figure 2B and Figure 3B), suggesting that trans-*N21* had better anti-microbial effect in the early stage of pathogen invasion. Considering the limitation of biopesticide Messenger or Illite in production, and the advantage of the N21 short peptide, we wondered whether N21 peptide could be more easily absorbed by the host plant and used as a succedaneum of Hpa1. To test this possibility, disinfected tomatoes were sprayed on the surface with the N21 peptide solution of 40 μg/mL, equal amount of Hpa1 protein expressed from *E. coli* and sterile water, left for 24 h under suitable conditions. Then the wounds were inoculated with hyphal blocks of *B. cinerea* and then blocks were removed at 24 hpi. The results showed that tomatoes treated with N21-PS presented the smallest lesions, only about 1.9 mm, compared to the non-treated tomatoes at 48 h of inoculation. However, diameters of infection zones of tomatoes treated with water was ~15.5 mm and a distinct layer of mildew developed in the diseased area. Tomatoes treated with same amount of Hpa1 protein exhibited distinctly depressed diseased area, but bigger than that in N21-PS-treated plants, about 6.9 mm (Figure 8A,B). Since the hyphal blocks were removed at 24 h, the onset of the disease was slower. At 96 h after inoculation, visible disease area within a small amount of mildew was observed in tomatoes treated with N21-PS. At the same time, tomatoes treated with water or Hpa1 protein all exhibited marked disease symptoms with more mildews, but there were still significant differences between three treatments (Figure 8A,B). All these results indicate that exogenous spray of N21 peptides exhibited a better bioavailability than that of the full-length Hpa1, and thus has higher disease resistance effect.

Apart from tomato, N21-PS or the same amount of Hpa1 protein expressed in *E. coli* were sprayed on the rice leaves before inoculating with the suspension of PXO99 of *Xoo*, separately. Rice leaves inoculated only with the suspension of PXO99 were treated as the positive control, and rice leaves treated with sterile water surface spray as the negative control. Then the plants were cultured in a constant temperature incubator at 25 °C with a RH of 80% after inoculation, and the incidence was recorded. The results showed that the symptoms of rice leaves going yellowing and dying were rather small, only had a lesion diameter of ~4.5 mm, when pretreated with N21 peptide solution, whereas the leaves pretreated with Hpa1 protein exhibited larger yellowing lesions, about 15.9 mm in lesion diameter. At the same time, the leaves treated only with suspension of PXO99 had the most severe disease symptoms, the diameter of lesions was ~51.8 mm (Appendix A). Measurement of relative content of *Xoo* also showed consistent results, which were significantly different among the three treatments (Appendix A). All these results indicated that both N21 and Hpa1 could increase the resistance of rice to *Xoo* and the N21 peptide had much better bioavailability than the full-length Hpa1 protein.

## 3. Discussion

We showed in this study that the most stable and shortest coiled-coil regions formed by 21 amino acid residues (3 heptads) encoding N21_Hpa1_ could induce HR in non-host tobacco using both proteins purified from *E. coli* and trans-*N21* tobacco. Trans-*N21* effectively enhanced resistance of tobacco to TMV and *Pcc*. The spraying of a N21 peptide solution effectively reduced the brown rot disease of peach, and this antifungal effect was much better than fungicides such as procymidone. Further, the N21 peptide delayed the occurrence of disease, reduced the disease severity, and improved the resistance of strawberries and tomatoes to *B. cinerea*. We also found that exogenous spray of N21 peptide promoted the growth of roots and increased of plant height in wheat, pepper, tomato and other plants, and induced drought tolerance in tobacco even more effective than Hpa1_Xoo_. These effects are consistent with previous studies for the effects of harpin proteins in promotion of plant growth, development, disease resistance and other beneficial phenotypes [1,2,6,7,36,40,41]. These results suggested that N21 enabled plants to obtain beneficial phenotypes, such as disease resistance, increased growth and drought tolerance, in vivo and in vitro. Our results indicated that coiled-coil N21 is an important functional unit of Hpa1_Xoo_.

Plants have a sophisticated immune system to respond to pathogens or microorganisms, including PAMP-triggered immunity (PTI) and effector-triggered immunity (ETI). These systems recognize the pathogens using immune receptors, which activate downstream defense responses, such as oxidative bursts, hypersensitive reactions, and the activation of MAPK and Ca^2+^ signaling [42,43,44,45]. To further understand the molecular mechanisms of induced resistance by N21 and full-length Hpa1_Xoo_, we examined the expression of a series of defense response genes. Notably, the defense-related genes *PR-1a*, *PR-1b* and *PR2*, especially *PR-1a*, were highly upregulated, which may underlie the disease resistance induced by coiled-coil N21. These genes are SA signaling-related genes, and phytohormones including JA, SA and ET are well known to be involved in plant defense responses [46,47,48]. SA plays pivotal roles in biotrophic pathogens to activate plant defense, and ET and JA primarily function in necrotrophic pathogens [42,49].

We also found that the ET synthesis-related gene *ACS2*, the JA synthetic genes *LOX1* and *AOC4*, and the SA-regulated genes *PR-1a*, *PR-1b*, *PR2* and *NPR1* showed higher expression in N21-expressing and Hpa1-expressing tobacco plants than that of in EV plants, which indicate that the overexpression of N21 or Hpa1_Xoo_ in plants might activate hormone-controlled signaling pathways to induce resistance mechanisms, such as growth promotion, disease and drought tolerance. Our results showed that the heights of trans-*N21* and trans-*Hpa1* tobacco plants were much higher than that of EV tobacco, and exogenous application of the N21 peptide solution enhanced the ability of drought tolerance of tobacco plants. Promotion of plant growth and stress resistance by harpins in vitro and in vivo have been reported previously in other plant species. For instance, overexpression of harpin genes or spray of a harpin solution was found to be able to promote plant growth via regulation of the ET-mediated signaling pathway [8,35]. Niu et al. (2019) have demonstrated that introduction of harpin_Xoc_-encoding gene *hrf2* in soybean could enhance the resistance of *Phytophthora sojae* via the upregulation of SA- and JA-dependent genes [50]. These studies, together with our results, suggest that SA and JA work synergistically to induce biotic or abiotic stress resistance in host plants.

The respiration burst oxidase genes *RBOHA* and *RBOHB* were upregulated in trans-*N21* and trans-*Hpa1* tobacco plants, which indicate that the ROS production in transgenic plants might have been increased to activate the expression of disease-resistant genes, such as *PR-1a*, *PR-1b* and *PR2*, and thus inhibiting the invasion of pathogens and increasing plant’s disease resistance. The expression levels of *PR-1a*, *PR-1b*, *PR2*, *LOX1*, *ACS1* and *RBHOA* genes were notably different between trans-*N21* and trans-*Hpa1* plants, indicating that, although the overexpression of N21 in plants or the exogenous addition of N21 peptide produced similar phenotypes to full-length Hpa1_Xoo_ in disease resistance, drought tolerance and growth promotion, trans-*N21* and trans-*Hpa1* have different functions in defense responses. N21 is just a small part of the Hpa1 protein (full length consists of 139 aa), and our previous work demonstrated an opposite function of the coiled-coil CC domain in the N-terminal and C-terminal of Hpal_Xoo_ [12]. In addition, Dong et al. has demonstrated both the N-terminal and C-terminal α-helices of Hpa1_Xoo_ mutation could induce reduction of the pathogenicity of *Xanthomonas oryzae* pv. *oryzae* [9], indicating that the 21 aa of C-terminal Hpa1_Xoo_ also play an important role in the pathogenesis of disease or other sides of pathogen. These might be the reasons for the differential expression of the defense-related genes in trans-*N21* and full-length transgenic tobacco.

In the present study, we demonstrated that the coiled-coil structure N21 in the N-terminal had similar function with Hpa1, with even stronger effect in inducing drought tolerance. Besides, we were surprised that the relative content of virus or bacteria in trans-*N21* tobacco were significantly lower than that in the trans-*Hpa1* plants in early stage of infection, and which tended to be similar in the later stage of infection. Furthermore, results of this study indicated that N21 peptide was more easily absorbed by plants than the full-length Hpa1 protein, and exhibited a better bioavailability, which supports the broad application prospects of this peptide as a promising succedaneum to Messenger or Illite or other biological pharmaceutical products, and it is possible to integrate with other products, such as engineered bacteria.

In summary, in this study we examined the biological function of the most stable and shortest three heptads N21 of Hpa1_Xoo_ in vivo and in vitro. The trans-*N21* tobacco plant exhibited improved resistance to TMV and *Pcc*, which was consistent with the results from trans-*Hpa1* tobacco. Treatment with a N21 peptide solution delayed the time of disease occurrence of *M. fructicola* and *B. cinerea*, promoted plant growth and drought tolerance in tobacco plants. N21 induced the expression of multiple plant defense-related genes, and had better bioavailability than the full-length Hpa1 protein. Our studies provide a basis for further development and use of proteins with similar structures.

## 4. Materials and Methods

### 4.1. Plant Materials, Pathogenic Strains, Pesticides and Growth Conditions

*Nicotiana tabacum* L. “Xanthi”, trans-*Hpa1* tobacco, EV tobacco, TMV, *Pcc*, *B. cinereal*, PXO99 and *M. fructicola* were maintained in the laboratory. Tobacco strains of expressing-*Hpa1_Xoo_* and empty vector, and *E. coli* strains expressing N21 in were obtained from Nanjing Agriculture University [12,13,32,51]. Trans-*N21* tobacco was obtained using the leaf-disc method after at least three generations of sub-culture. 35S promoter of Cauliflower mosaic virus (CaMV)_was used to drive the expression of N21 and Hpa1 in the transgenic tobacco. The N21 peptide was synthesized using the *F-moc* solid-phase peptide synthesis method (Nanjing Kingsrui company, China). Synthetic peptides were packaged in tubes of 4 mg each and stored as freeze-dried powder at −80 °C.

Strawberries (“Fengxiang”) and tomatoes (“Jinguan” no. 5) came from experimental and demonstration bases of Jiangwang in Yangzhou, Jiangsu, China. Peach plants (“Hujingmilu”) were from peach experimental and demonstration bases at Yangshan, Wuxi, China. Seeds, such as pepper (“Huayu” 8819), melon (“Yangyan”), cucumber (“Jinyang” of new no. 4), wheat (“Ningmai”13), tomato (“Dehua”), procymidone (Sumitomo Chemical Industry Corporation of Japan, 50% wettable powder, Shanghai, China) and carbendazim (Taicang Agricultural Pharmaceutical Factory Co. LTD of China, 70% wettable powder, Suzhou, China) were purchased from agricultural stores in China.

*Nicotiana tabacum* L. “Xanthi”, trans-*N21* tobacco, trans-*Hpa1* tobacco and EV tobacco plants were potted in a greenhouse at 25 °C with 80% humidity for 7–8 weeks (approximately 6–7 leaf stage).

### 4.2. Acquisition of N21-Transgenic Tobacco

The expression vector was transformed into EHA105 of *Agrobacterium tumefaciens* using a freeze-thaw method to obtain recombinant strains. The antibiotic-resistant single colony was cultured in YEB media with 50 μg/mL of kanamycin (km) (Diamond, A100408-0005, Sangon Biotech, Shanghai, China) and 25 μg/mL of chloramphenicol (Cm) (Diamond, A100230-0010, Sangon Biotech, China) to the logarithmic phase, centrifuged at 4000 r/min for 10 min, washed and resuspended in 20 mL Murashige & Skoog medium (MS) through standard procedure. The selected leaves of tobacco seedlings were completely spread out and disinfected, and the leaf plates were taken with a sterilizing perforator. The leaf disc was placed into the bacterial suspension using sterile forceps and vortexed 30 s to ensure that the transforming bacterial solution fully contacted the wound site of the leaf tissue. The discs were placed on sterile filter paper to allow the excess bacteria on the surface to dry, and placed on an MS co-culture medium (MS + 1 mg/L 6-BA) for 3–4 days in the dark at 25 °C. The transformed leaf discs were transferred to the differentiated media (MS + 100 mg/L km+ 500 mg/L Carbenicillin (Cb) + 1 mg/L 6-BA) for 5–7 weeks at 25 °C with a relative humidity (RH) of 66%. When the buds grew to greater than 2~3 cm, the buds were cut and transferred to the rooting medium (1/2 MS + 100 mg/L km + 0.2 mg/L IAA). The roots were grown for 2–3 weeks, and the seedlings were transplanted into a pot and cultured in a greenhouse (25 °C, 80% RH). gDNA of transgenic tobacco was extracted using the AxyPrep Multisource Genomic DNA Miniprep Kit (Axygen, Tewksbury, MA, USA). RNA was extracted using the PureLink^TM^ RNA Mini Kit (Invitrogen, Cat no. 12183018A, Nanjing, China). PCR and semi-quantitative RT-PCR detection, and sequencing were performed to verify the correctness of the target gene (*N21*) inserted into the transgenic tobacco. Transgenic seeds were propagated continuously to T3 generations by km selection, PCR and semi-quantitative RT-PCR. The primers used are listed in Appendix A.

### 4.3. Protein Activity Assay in Trans-N21 Tobacco Plants

Forty-five-day fresh leaves of trans-*N21* tobacco were cut and immediately ground into a powder using liquid nitrogen. The samples were placed into 1 mL of plant protein extraction buffer (50 mM Tris-HCl (pH 7.0), 10 mM MgCl_2_, 1 mM EDTA, 5 mM DTT, 5% PVP, 10% glycerine) and 10 μL PMSF (100 mM), shocked for 10 min, and incubated for 3 h at 4 °C. The samples were centrifuged at 12,000 r/min for 20 min at 4 °C, and the supernatant was taken. The supernatant was dried then dissolved in 500 μL sterile water, and measured protein concentration via Bradford protein concentration assay kit (Beyotime, P0006) for the detection of protein activity. A small hole was created in the lower epidermis of a fully expanded leaf of Xanthi tobacco, and the prepared protein solution was injected into the hole using a 1-mL needle. The plants were cultured in a 16 h/8 h light/dark cycle at 25 °C for 24 h to observe the results. Proteins of EV tobacco and Xanthi tobacco were extracted as negative controls. Proteins of N21 and Hpa1 directly expressed by *E. coli* BL21 cells were used as positive controls. The experiment was repeated three times with the same results.

### 4.4. Determination of Resistance of Trans-N21 Tobacco to TMV and Pcc

TMV was inoculated via friction inoculation. A small amount of quartz sand (400 mesh) was scattered on the leaves. A TMV suspension (10 μL) was evenly dripped onto the leaves of trans-*N21* tobacco, and inoculated via gently rubbing the leaves with fingers. After inoculation, the leaves were gently rinsed with water, and the experimental plants were cultured in an isolated greenhouse at 25 °C for 36–72 h. EV and trans-*Hpa1* tobacco were used as controls in each treatment. Ten plants were inoculated in each transgenic strain and three leaves of the same leaf age were inoculated in the middle of each plant. The experiment was repeated three times.

Fully unfolded leaves of trans-*N21* plants were dipped in the suspension of *Pcc* at a concentration of 1.0 × 10^7^ CFU/mL and cultured at 25 °C for 12–16 h. Resistance was analyzed with reference to Ger et al. [34]. EV and trans-*Hpa1* tobacco were used as controls. The experiment was repeated three times, and each experiment had three replicates.

### 4.5. Determination of Resistance of N21 Peptide to M. Fructicola and B. cinerea

Peach surfaces were disinfected with 75% alcohol, cleaned with sterile water three times, and dried in a cool place. The peaches were treated with a surface spray of sterile water, an N21-PS (40 μg/mL) and 1000 times diluted procymidone. The treated plants were placed in an illumination incubator for 13 h at 25 °C with a relative humidity of 50% for 24 h. Inoculated blocks of *M. fructicola* on treated peaches with needle wound at 25 °C for 3 days. The experiment was repeated three times, and each experiment had three replicates.

Strawberries and tomatoes were disinfected as described above, and treated separately with sprays of sterile water, the N21-PS (40 μg/mL) and carbendazim diluted 800 times. Conidial suspensions (5 × 10^5^ spores/mL) or blocks of *B. cinereal* were inoculated on the wounded surface of strawberry and tomato plants after different treatments for 24 h, and the disease symptoms were observed daily. The experiments were repeated with three independent materials in three replicates.

### 4.6. Determination of Growth-Promoting Effect of N21 Peptide

For the growth promotion of tomato plants, 30 tomato seeds were disinfected with 75% alcohol for 5 min, dipped in 30% sodium hypochlorite for 30 min and cleaned with sterile water 5 times. After germination, planted the seeds in the pots at 25 °C in an incubator with a 13 h/11 h light/dark cycle and divided into two groups: normal watering or sprayed with a N21-PS (40 μg/mL) for 5 mL every 10 days. After 45 days, the length of roots, fresh weight and plant height were measured.

For the growth promotion of seeds of different plants, including tomato, pepper, cucumber, melon and wheat, the same disinfection was performed for the seeds. The seeds were separately dipped into an N21-PS (40 μg/mL) or water at 4 °C for 3 h then transferred to the incubator with 13-h light at 25 °C for 5 days. The root length was recorded. Finally, the seeds were transplanted to pots and treated with the N21 solution or water. Plant height was recorded on the tenth day.

### 4.7. Determination of Drought Tolerance Induced by N21 Peptide

The roots of 6-week-old Xanthi tobacco were watered with a 10% PEG6000 solution daily (treated 2 days), and the plant surface was sprayed with a solution of 80 μg/mL N21-PS, the same amount of Hpa1 protein or sterile water. Survival rate was measured with the seedling of 7-day-old Xanthi tobacco under the same treatment. Hpa1 protein was expressed from *E. coli* and the protein concentration was measured using the standard Bradford Protein Assay Kit (Beyotime P0006). Pictures were taken every 6 h. For germination rate assay, seeds of different transgenic lines were germinated on MS media containing 200 mM mannitol [52]. Seeds germinated on MS media were used as controls. Germination rate were calculated after 21 days. Measurement of relative water content was according to Sharma et al. [52]. The leaves showed dehydration symptoms after another 48 h treatment was cut for drought-related gene expression analysis. The experiments were repeated three times, and each experiment had three replicates.

### 4.8. Bioactivity Assay of Hpa1 and the N21 Peptide

Tomatoes were surface-disinfected with 75% alcohol, washed with sterile water 3 times, and dried at a cool place. The tomatoes were then divided into three groups, and each group with 3 tomatoes. One group was treated with 3 mL of 40 μg/mL N21-PS; the second group was treated with the same amount of the Hpa1 protein expressed from *E. coli*; the third group treated with 3 mL of sterile water as the negative control. Then, the tomatoes were placed in a 12 h/12 h light/dark cycle incubator at 25 °C for 24 h. Tomatoes were taken out, made wounds and inoculated with hyphal blocks of *B. cinera*. After inoculation, they were cultured in an incubator at 25 °C with the relative humidity of 50%, and incidence recorded and photographed every 48 h. The experiments were repeated three times, and each experiment with three replicates.

### 4.9. Quantitative Real-Time PCR Assays

To analyze the expression of genes associated with the plant defense response in different transgenic tobacco strains, total RNA was extracted using the PureLink^TM^ RNA Mini kit (Invitrogen, Cat no. 12183018A, USA). Bio-Rad was used for qRT-PCR analysis. The *EF-1α* gene was used as the internal control. The experiment was repeated three times, and each experiment included three replicates. The primers used in this paper are shown in Appendix A.

## Figures and Tables

**Figure 1 ijms-22-00203-f001:**
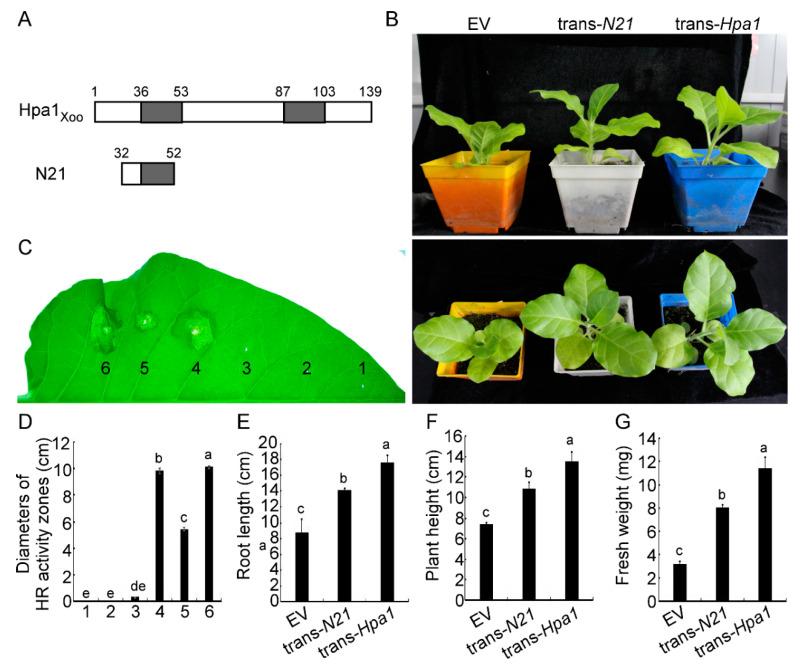
Growth measurement and protein activity assay of trans-*N21* tobacco plants in T3 generation. (**A**) Hpa1-N21_Xoo_ was composed of three heptads repeats, which included part of the α-helix region. Grey area represents α-helix regions. (**B**) The trans-*N21*, trans-*Hpa1* and EV tobacco plants were placed in a greenhouse at 25 °C (RH 80%) with alternating light and dark at 16 h/8 h, and photos were taken 45 days after planting. EV represented transgenic empty vector (*pBI121*) tobacco. (**C**) Proteins of trans-*N21* tobacco induced hypersensitive response (HR) in Xanthi tobacco. Treatments were infiltrated (from right to left): 1. Water; 2. proteins of Xanthi tobacco; 3. proteins of EV tobacco; 4. proteins of N21 expressed by *Escherichia coli* BL21 cells; 5. proteins of trans-*N21* tobacco of T3 progeny; 6. proteins of Hpa1 expressed by *E. coli* BL21 cells. All protein contents were 5 mg. (**D**) Diameters of HR activity zones were measured at 24 h post inoculation. Measurements of (**E**) root length, (**F**) plant height and (**G**) fresh weight were taken at 45 days after planting. Error bars represent the standard deviation and letters represent significant differences (Duncan’s new multiple range test, *p* < 0.05). Empty vector expressing tobacco strains (EV) represented transgenic empty vector (*pBI121*) tobacco.

**Figure 2 ijms-22-00203-f002:**
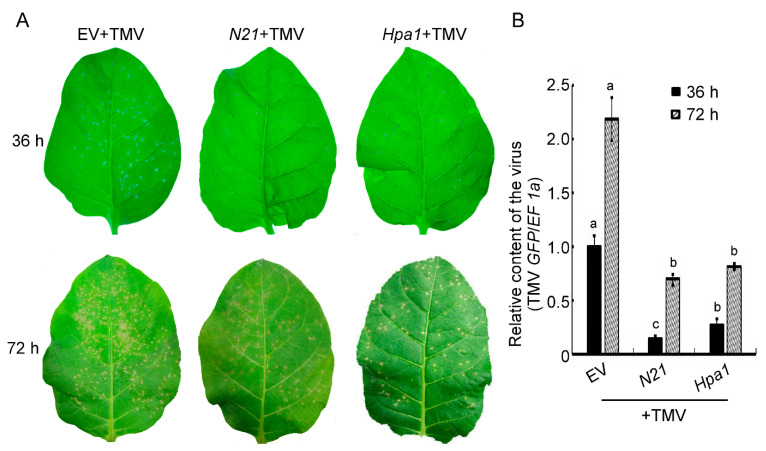
Trans-*N21* tobacco showed enhanced resistance to tobacco mosaic virus (TMV). (**A**) Disease symptoms produced on 8-week different transgenic tobacco strains treated with a TMV suspension (10 μL) via friction vaccination at 36 or 72 h post inoculation (hpi). (**B**) Relative content of virus in the diseased leaves of EV, *N21* and *Hpa1* tobacco were measured by quantifying TMV *GFP* relative to tobacco genomic *EF 1α* at 36 or 72 hpi. Error bars represent the standard deviation and letters represent significant differences (Duncan’s new multiple range test, *p* < 0.05). EV, *N21*, *Hpa1* represented transgenic empty vector tobacco, trans-*N21* tobacco and trans-*Hpa1* tobacco, separately.

**Figure 3 ijms-22-00203-f003:**
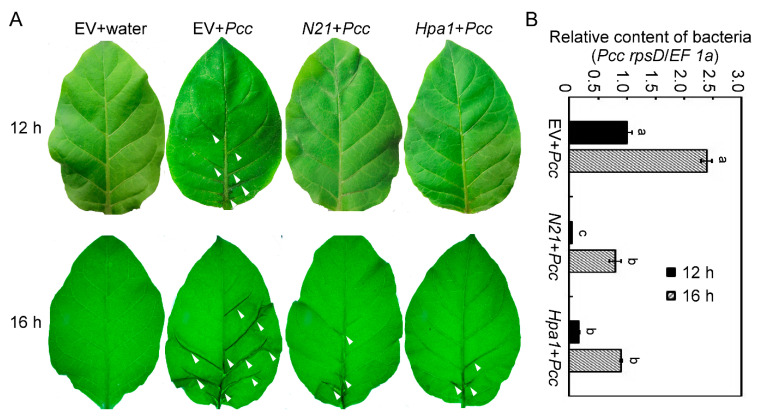
Trans-*N21* tobacco has enhanced resistance to *Pectobacterium carotovora* subsp. *carotovora* (*Pcc*). (**A**) Disease symptoms produced on 8-week different transgenic tobacco strains dipped in suspensions of *Pcc* at 12 or 16 h post inoculation (hpi). Treatment with water on EV tobacco was the negative control. The white arrow indicated the affected area. (**B**) Relative contents of bacteria in the diseased leaves of EV, *N21* and *Hpa1* tobacco were measured by quantifying *Pcc rpsD* relative to tobacco genomic *EF 1α* at 12 or 16 hpi. Error bars represent the standard deviation and letters represent significant differences (Duncan’s new multiple range test, *p* < 0.05). EV, *N21*, *Hpa1* represented transgenic tobacco expressing the empty vector, N21or Hpa1protein, separately.

**Figure 4 ijms-22-00203-f004:**
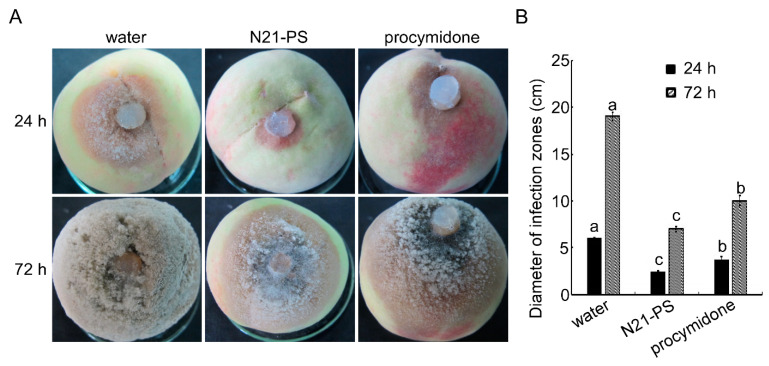
N21 peptide induces host resistance to peach brown rot. (**A**) Peaches were disinfected with 75% alcohol, washed with sterile water and dried in a cool place. They were divided into three groups: the first group was sprayed with sterile water; the second group was sprayed with a N21 peptide solution (N21-PS) (40 µg/mL); the third group was sprayed with 1000 times diluted procymidone. Peaches were cultured in an incubator with 13 h continuous light at 25 °C with a relative humidity of 50% and removed after 24 h. Hyphal blocks of *M. fructicola* were inoculated on the wounded surface of peaches with different treatments. Incidences were observed every 24 h under the same environmental conditions, and photos were taken at 24 and 72 h post inoculation (hpi). (**B**) Measurement of the diameter of infection zones at 24 and 72 hpi. Error bars represent the standard deviation and letters represent significant differences (Duncan’s new multiple range test, *p* < 0.05).

**Figure 5 ijms-22-00203-f005:**
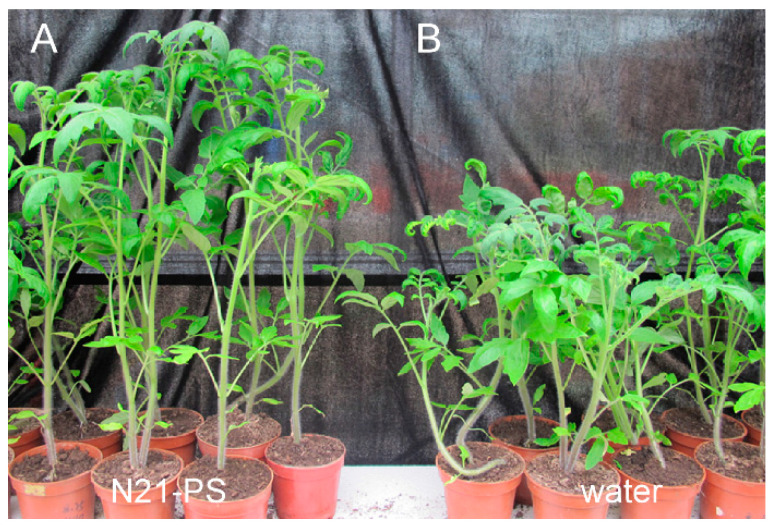
N21 peptide promotes the growth of tomato plants. Thirty tomato seeds were disinfected with 75% alcohol, soaked with 30% sodium hypochlorite for 30 min and washed with sterile water 5 times. These seeds were placed in sterile water to accelerate germination at 25 °C. After the seeds germinated, they were divided into two groups: one group was watered with a spray of the N21 peptide solution (N21-PS) (40 µg/mL) every ten days (**A**), the other was watered every ten days (**B**). The growth of tomato seedlings was observed and photographed at 45 days.

**Figure 6 ijms-22-00203-f006:**
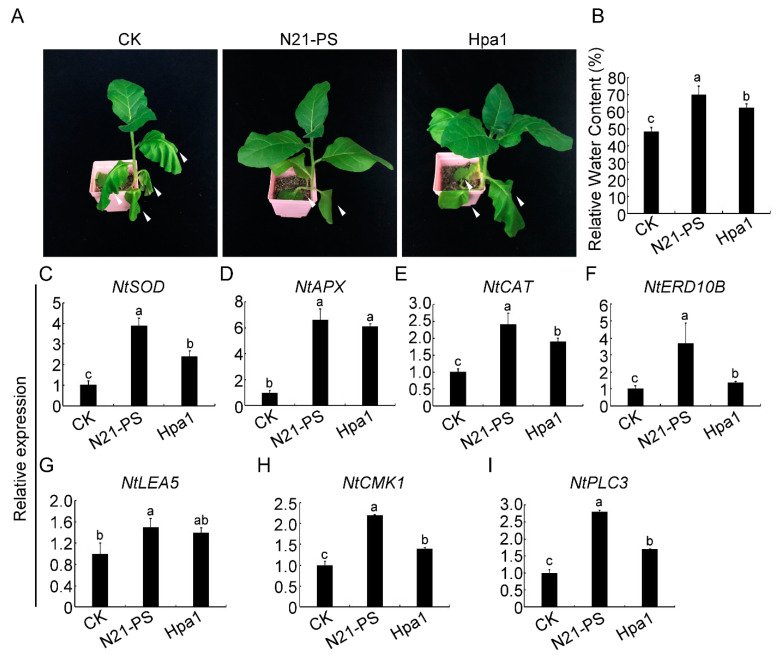
N21 peptide improves the drought tolerance of tobacco plants. (**A**) Drought stress of tobacco was treated with a 10% polyethylene glycol 6000 (PEG6000) solution. Nine Xanthi tobacco plants were divided into three treatment groups. The roots of plants were irrigated with 10% PEG6000 for 48 h, and sprayed on the surface with sterile water (the left plant), 80 µg/mL N21 peptide solution (N21-PS) (the middle plant) and the same amount of Hpa1 expressed in *Escherichia coli* (*E. coli*) (the right plant) every day. Photos were taken at 48 hpi. The white arrow indicated the arid area. (**B**) Different treatment effects were observed by estimating relative water content (%). Error bars represent the standard deviation and letters represent significant difference (Duncan’s new multiple range test, *p* < 0.05). The relative expression of stress-responsive genes of (**C**) *NtSOD*; (**D**) *NtAPX*; (**E**) *NtCAT*; (**F**) *NtERD10B*; (**G**) *NtLEA5*; (**H**) *NtCMK1*; (**I**) *NtPLC3* was measured by qRT-PCR under different treatments. Error bars represent the standard deviation and letters represent significant differences (Duncan’s new multiple range test, *p* < 0.05).

**Figure 7 ijms-22-00203-f007:**
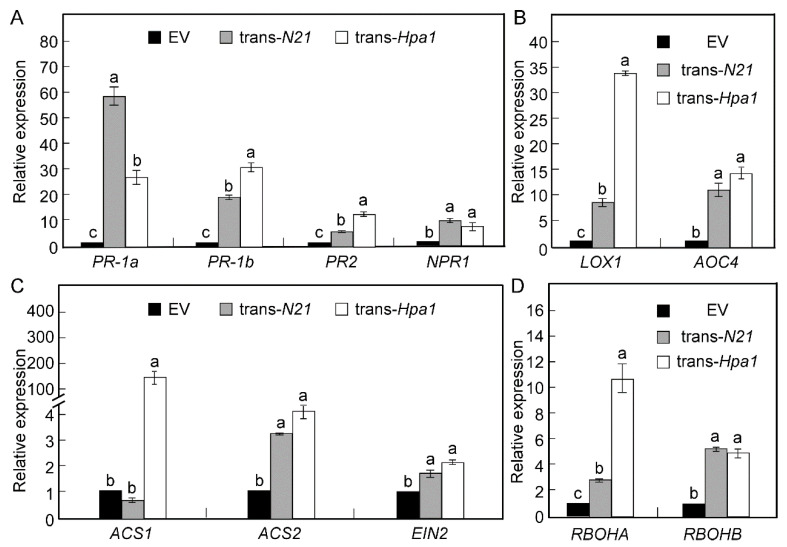
Expression analysis of plant defense-related genes in different transgenic tobacco. Total RNA was separately extracted from fully unfolded leaves in 6-week tobacco strains expressing pBI121 vector (EV), trans-*N21* and trans-*Hpa1*. cDNA was reversed transcribed. (**A**–**D**) The relative expression of four salicylic acid (SA) signaling-related genes (*PR-1a*, *PR-1b*, *PR2*, *NPR1*) (**A**), two jasmonic acid (JA) biosynthesis genes (*LOX1*, *AOC4*) (**B**), three ethylene (ET) synthesis and signaling genes (*ACS1*, *ACS2*, *EIN2*) (**C**), and two reactive oxygen species (ROS) synthesis-related genes (*RBOHA*, *RBOHB*) (**D**). *EF-1α* was used as a reference gene. Standard deviation (±SD) was calculated from three repeated experiments, and lowercase letters indicate statistically significant differences (Duncan’s new multiple range test, letters mean *p* < 0.05).

**Figure 8 ijms-22-00203-f008:**
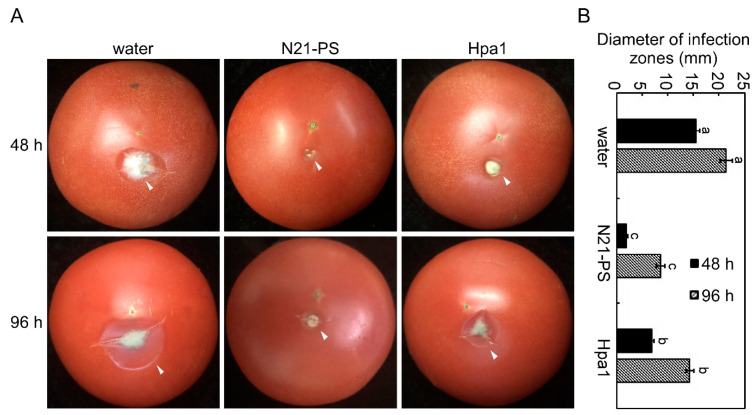
The N21 peptides of *Xanthomonas oryzae* pv. *oryzae* showed a better plant bioavailability than that of the full-length Hpa1 protein. (**A**) Tomatoes were disinfected with 75% alcohol, washed with sterile water 3 times and dried. Then plants were treated with 3 mL of 40 μg/mL N21-PS, the same amount of Hpa1 protein expressed from *E. coli* or sterile water, separately, in an incubator with a 12 h/12 h light/dark cycle for 24 h. Then hyphal blocks of *B. cinera* were inoculated on the wounds and put them in an incubator with 50% relative humidity (RH). The incidence was recorded and photographed every 48 h. White arrows indicate the area of the disease. N21-PS represents N21 peptide solution. *E. coli* stands for *Escherichia coli*. (**B**) Diameter of infection zones was measured at 48 or 96 h post inoculation (hpi). Standard deviation (±SD) was calculated from three repeated experiments, and lowercase letters indicate statistically significant differences (Duncan’s new multiple range test, letters mean *p* < 0.05). hpi represents hours post inoculation.

**Table 1 ijms-22-00203-t001:** Resistance levels of transgenic tobacco to TMV.

Tobacco Strains	Number of Lesions ^a^	Number of Lesions ^b^
EV	112 ± 11.0	462 ± 34
trans-*N21*	33 ± 5.0 *	115 ± 25 *
trans-*Hpa1*	31 ± 8.0 *	88 ± 23 *

Number of lesions on different transgenic tobacco inoculated with tobacco mosaic virus (TMV) at 36 (^a^) and 72 h post inoculation (hpi) (^b^). ±SD was calculated from three repeated experiments and asterisks indicate statistically significant differences (Duncan’s new multiple range test, * means *p* < 0.05). EV represents tobacco strains expressing the pBI121 vector.

**Table 2 ijms-22-00203-t002:** Resistance of N21 peptide to grey mould.

Plants	Treatment	Disease Time ^a^ (d)	Diameter of Lesions ^b^ (cm)	Inhibition Rate ^c^ (%)
strawberry	N21-PS	4	2.4 ± 0.4 *	31.4 ± 4.1
	carbendazim	>4	1.6 ± 0.2 *	54.3 ± 1.7
	water	2	3.5 ± 0.3	
tomato	N21-PS	3	3.9 ± 0.5 *	48.6 ± 1.6
	carbendazim	2	4.5 ± 0.3 *	40.0 ± 2.1
	water	2	7.5 ± 0.8	

Thirty strawberry or tomato plants were disinfected with 75% alcohol, then washed with sterilized water 3 times and dried. The plants were divided into three groups: sprayed with the N21 peptide solution (N21-PS) (40 μg/mL), carbendazim diluted 800 times (~0.88 mg/mL) or sterilized water. Plants were put in an incubator for 24 h with a 13 h/11 h light/dark at 25 °C. Conidial suspensions (5 × 10^5^ spores/mL) of *B. cinereal* were inoculated on the strawberries or hyphal blocks of *B. cinereal* were inoculated on the tomatoes on a needle-wounded surface, cultured in an incubator with a humidity of 50% at 25 °C, and the disease symptoms were observed every 24 h. ^a^. The disease time recorded by different treatments. ^b^. Measurement of lesion diameters on the fourth day of different treatments. ^c^. Statistical analysis of inhibition rate compared to treatments of water. Standard deviation (±SD) was calculated from three repeated experiments and asterisk indicates statistically significant differences (Duncan’s new multiple range test, * means *p* < 0.05).

**Table 3 ijms-22-00203-t003:** Promoting effect of N21 peptide on growth of tomato plants.

Treatment	Plant Height (cm)	Fresh Weight (g)	Root Length (cm)
N21-PS	49.0 ± 0.2 *	20.1 ± 0.3 *	11.7 ± 0.3 *
water	43.3 ± 0.3	16.3 ± 0.5	10.9 ± 0.1

Thirty tomato seeds were disinfected with 75% alcohol immersed for 5 min, and dipped into 30% sodium hypochlorite solution for 30 min, then washed with sterilized water 5 times. After seed germination, plants were placed in an incubator with the cycle of 13 h/11 h light/dark at 25 °C and sprayed with sterile water or the N21 peptide solution (N21-PS) (40 μg/mL) every ten days. Plant height, fresh weight and root length were measured and photographed after 45 days. Standard deviation (±SD) was calculated from three repeated experiments and asterisk indicates statistically significant differences (Student’s *t* test, * means *p* < 0.05).

**Table 4 ijms-22-00203-t004:** Promoting effect of N21 peptide to seeds of different plants.

	Treatment	Tomato	Pepper	Cucumber	Melon	Wheat
Root length ^a^ (cm)	N21-PS	5.73 ± 0.25 *	7.90 ± 0.44 *	4.80 ± 0.26 *	4.10 ± 0.26 *	8.60 ± 0.62 *
	water	5.13 ± 0.21	6.33 ± 0.45	3.17 ± 0.23	3.67 ± 0.15	8.20 ± 0.30
Plant height ^b^ (cm)	N21-PS	9.10 ± 0.20 *	13.37 ± 0.45 *	16.00 ± 0.44 *	9.17 ± 0.47 *	35.30 ± 0.40 *
	water	8.10 ± 0.30	10.33 ± 0.25	13.23 ± 0.35	8.57 ± 0.21	31.40 ± 0.82

Seeds of tomato, pepper, cucumber, melon and wheat were disinfected with 75% alcohol for 5 min, soaked with 30% sodium hypochlorite for 30 min, and finally rinsed with sterilized water 5 times. Seeds of each variety were divided into two groups: dipped in a N21 peptide solution (N21-PS) (40 μg/mL) or sterilized water at 4 °C for 3 h then transferred to an incubator at 25 °C with a cycle of 13 h/11 h light/dark. ^a^. Root length of different plants were measured after 5 days. Standard deviation (±SD) was calculated from three repeated experiments and asterisk indicates statistically significant differences (Student’s *t* test, * means *p* < 0.05). ^b^. Plant height of different plants transplanted into the basins were measured after 10 days. ±SD was calculated from three repeated experiments and asterisk indicates statistically significant differences (Student’s *t* test, * means *p* < 0.05).

## Data Availability

All the data is actually available.

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
