# Peer review of "Coiled-Coil N21 of Hpa1 in Xanthomonas oryzae pv. oryzae Promotes Plant Growth, Disease Resistance and Drought Tolerance in Non-Hosts via Eliciting HR and Regulation of Multiple Defense Response Genes"

_ijms, 2020, doi:10.3390/ijms22010203_

Round 1

Reviewer 1 Report

MS ijms-1055319, entitled Coiled-coil N21 of Hpa1 in Xanthomonas oryzae pv. oryzae promotes plant growth, disease resistance and drought tolerance in non-hosts via eliciting HR and regulation of multiple defence response genes" has been revised according to reviewer recommendations, new results have been added, and methods have been more clearly explained. Therefore the revised MS has been largely improved and it is almost ready for its publication. However, there are still some style and edition mistakes that should be corrected.  Some of the edition changes are indicated in the attached MS revised form. Nevertheless, English style should be thoroughly polished in the final version. I recommend a careful read through once the changes have been done, before the final submission of the revised MS.

Reviewer 2 Report

Article Title: Coiled-coil N21 of Hpa1 in Xanthomonas oryzae pv. oryzae promotes plant growth, disease and drought resistance in non-hosts via eliciting HR and regulation of multiple defense response genes

Manuscript ijms-1055319

The investigation has been greatly improved. A considerable amount of additional experimental work has been done, and the data presentation and literature analysis have been also improved. The authors properly addressed all my comments. I am glad to recommend this manuscript for publication after minor corrections. Please find minor comments below:

  • Title: Correct “defence” to “defense”. Check throughout the Ms.

Introduction.

  • p. 1, line 37 “Hpa1 ….is encoded by 139 amino acids”.

“encoded” is used when talking about a gene. Therefore, correct to “contains” or “consists of” etc.

  • p. 2, line 65. Correct “Wang and colleague…” to “ Wang and colleagues…”
  • p. 2, line 66. Did you mean abolished?
  • p. 2. line 74-75. Correct “were developed as biopesticides in the early stage because of its promoting of plant growth, increasing crop  yield,  inducing  plant  disease  resistance,  non-toxicity,  lack  of  residue  and  environmentally  friendliness.” TO “were developed as biopesticides in the early stage because of their promoting effects on plant growth, crop  yield,  plant  disease  resistance,  non-toxicity,  lack  of  residue  and  environmental  friendliness.” It is also not clear what is “in the early stage” here? Maybe delete it? In addition, check the phrase “lack of residue”. Sounds a bit weird.  
  • p. 2, line 80-81. I recommend delete this “and its characteristics of short peptide”. Is is not clear here.
  • p.2, line 82-83. Correct “To test this conjecture, we performed a series of studies and the results suggested that..” TO “Results of the present work suggested that …”. English problems.
  • p. 2, line 88. Correct “…enhancement of  drought  tolerance  that  is  even  better  than  the  full  length.” TO “…enhancement  of  drought  tolerance  that  is  even  better  than  that of the  full-length protein”
  • p.2, line 89. Correct “and N21 peptide had better bioavailability” TO “and had a better bioavailability”.
  • Figure 1 legend, p. 3, line 113. Improve title of the legend (1st sentence). This title is grammatically incorrect and does not reflect what has been dowe.
  • p.4, line 120. Delete “E. coli”
  • p.4, line 132. “N21- and Hpa1-expressing tobacco” Should be in italics. Check for this kind of mistakes throughout the Ms. When N21- and Hpa1 are mentioned as transgenes/genes, then italics.
  • p.5, line 149. Fig.2 Correct “enhance” to “enhanced”.
  • p.5, lines 161-167. Correct “However, we were able to detecte the bacteria using quantitative PCR in N21- and Hpa1- transgenic tobacco, and found that the…” TO “However, we were able to detect the bacteria using quantitative real-time PCR (qRT-PCR) in the N21- and Hpa1-transgenic tobacco and found that the…”.
  • p.5, lines 161-167. Correct “became significantly watery” and “bigger differences”. English problems.
  • p.5, lines 161-167. Correct “The results of the measurement  of  bacteria  content  were  in  accordance  with  it  (Fig.  3B).” TO “The results of the measurement  of  bacteria  content  showed a similar trend  (Fig.  3B).”
  • p.9, line 294. Correct “In addition, significant higher expression…” To “A significantly higher expression…”
  • p.9, line 296-29. Correct “was observed in the plants treated N21-PS and Hpa1 protein, relative to the water-treated plants, with highest expression in  N21-PS treatment (Fig. 6C-I). All results indicated that the N21 peptide could improve the drought  tolerance of tobacco plants even better than the full-length effect.”

TO “was observed in the plants treated with the N21-PS and Hpa1 proteins, relative to the water-treated plants, with the  highest expression in  the N21-PS treatment (Fig. 6C-I). All results indicated that the N21 peptide could improve the drought tolerance of tobacco plants even better than the full-length Hpa protein.”

  • p.12, line 356. Correct “at 24  h  post  inoculation.” To at 24 hpi”
  • p.12, lines 363-367. Correct TO “At the same time, tomatoes treated with water or Hpa1protein exhibited  **marked  disease  and  mycelium**(THIS PHRASE IS NOT CLEAR),  but  there  were  still  significant differences between the three treatments (Fig. 8A and 8B). All these results indicated that exogenous spray of N21 peptide  exhibit  a better  bioavailability  than  that of the  full-length  Hpa1,  and thus  has  higher  disease  resistance effect.
  • Fig.8 Title. Correct something like this “The N21 peptide of Xanthomonas oryzae pv. oryzae  showed  a better  plant bioavailability  than  that of the  full-length  Hpa1 protein.”
  • Discussion: Correct “Furthermore, we have proved that N21 was more easily absorbed by plants  compared  with  the  full-length  protein  and  had  better  bioavailability,  which  supports  the broad application prospects of this peptide as a promising succedaneum to Messenger or Illite or  other biological pharmaceutical products, and it is possible to integrated with other products, such  as engineered bacteria.” TO

“Furthermore, results of this study indicated that N21 peptide was more easily absorbed by  plants  than  the  full-length  Hpa1 protein,  and  exhibited a better  bioavailability,  which  supports  the broad application prospects of this peptide as a promising succedaneum to Messenger or Illite or  other biological pharmaceutical products, and it is possible to integrate with other products, such  as engineered bacteria.”

Author Response

This manuscript is a resubmission of an earlier submission. The following is a list of the peer review reports and author responses from that submission.

Round 1

Reviewer 1 Report

Article Title: Coiled-coil N21 of Hpa1 in Xanthomonas oryzae pv. oryzae promotes plant growth, disease and drought resistance in non-hosts via eliciting HR and regulation of multiple defence response genes

Manuscript ijms-924635

This paper reports that expression of a N21 peptide of Xanthomonas oryzae in tobacco and spraying leaves of different plants with the N21 peptide solution promote plant growth,  fungal and viral disease resistance, and drought stress tolerance. This Ms adds new important information on using pathogen-derived proteins for plant protection. Therefore, the topic of this manuscript (Ms) is interesting and relevant for IJMS due to its potential for the development plant treatment technologies. The authors also state that the approach is more promising than other known biopesticides based on harpin solutions due to a higher bioavailability of N21. There is a number of essential critical remarks related to data presentation and experiments in the Ms that should be corrected or performed before publication. Importantly, in my view, several additional experiments are required to improve the manuscript.

In conclusion, I think that this manuscript should be resubmitted after the additional experimental procedures and data presentation improvements are made. After that, I would recommend the manuscript for publication.

Specific comments to authors.

  • The most interesting and novel issue in this Ms is that it appears that N21 peptide presents better bioavailability for plants that than other known biopesticids based on harpin solutions. The authors mentioned this in Discussion. However, there is no clear evidence for N21 better bioavalialbilty. Please perform and discuss additional experiments to assess the N21 peptide bioavailability for plants when sprayed on the plant leaves and for transgenic plants. Here would be helpful the recommended below specific additional experiments for Figures 1, 3, 4,5, and 6 (the shown effects should be assessed quantitatively). Otherwise there is no evidence for a better bioavailability of N21 than at least Hpa1 and would be a lack of novelty in the MS (in literature, it has been already known on hairpin protein protective effect against plant diseases and drought).
  • Please compare the newly obtained results with the known in the literature hairpin bioinsecticides data and discuss the new data. It is important to mention on the bioavailability in the Introduction and in Abstract (if you would present some bioavailability studies).
  • Figure 1 B. Please assess the growth quantitatively (add either a graph or a table – e.g. plant fresh weight, height, root length).
  • Figure 1 C. Assess the effect quantitatively (e.g. measure the diameters of HR activity zones). Add the numbers 1,2,3,4,5,6 to the explanation in the Figure legend.
  • Figure 3 and Figure 4. Assess the effect quantitatively (e.g. the percentage of leaves with visual signs of infection in Fig. 3; the diameter of infection zones and/or percentage of infected peaches in Fig. 4).
  • Figure 5. Please provide a better picture (the plants should be organized better on the picture and A, B parts should be given or the plants should be signed directly on the photograph). It is also necessary to tie up the plants to show the readers whether there are difference in plant height and phenotype.
  • Figure 6. Why the tobacco drought resistance was assessed using PEG and not by retention of watering? Growing plants without watering is a much more natural drought induction condition. I recommend that the drought stress tolerance should be assessed by water withdrawal. Furthermore, it is important that the drought protection effect of N21 should be assessed quantitatively. In this case, it is better to assess drought tolerance on much younger tobacco plants by calculating plant survival rates per pot after drought.
  • Please explain why did you not test the plant resistance to Xanthomonas oryzae pv. Oryzaeis after spraying with N21 and/or for transgenic plants?
  • Please correct title (English problems). For example to : “Coiled-coil N21 of Hpa1 in Xanthomonas oryzae pv. oryzae promotes plant growth, disease resistance, and drought tolerance in non-hosts via eliciting HR and regulation of multiple defence response genes”.
  • Abbreviations HR, EV, TMV, Pcc and so on should be explained in all the Figure legends. TMV should be also explained when is given in the Introduction first time. Please also add a list of abbreviations at the end of Ms.
  • Please clarify, what promoter has been used to drive N21 and Hpa expression in the transformed tobacco?

Reviewer 2 Report

Manuscript ID: ijms-924635, entitled “Coiled-coil N21 of Hpa1 in Xanthomonas oryzae pv. oryzae promotes plant growth, disease and drought resistance in non-hosts via eliciting HR and regulation of multiple defence response genes”, shows the comparison of the activity of a small fragment and the full-length, whole Harpin Hpa1 protein. Harpins are virulence factors, and elicitors of defense typically injected by the TSS of plant pathogens, and many results already demonstrate that different domains of several of these harpin proteins are able to elicit broad range defense. In this piece of work, authors show that a 21 residue peptide (3 heptad forming an N-terminal coiled-coil domain), is also able to elicit defense in several host and against various pathogens. Authors also show induced expression of defense genes acting in several defense pathways. In this sense, the results might be promising about the use of the smallest possible peptide as a possible bio-pesticide, with additional growth-promoting effects (which are also typical features of harpins). However, there a series of flaws that should be corrected before the manuscript could be published.

Mayor's drawbacks are that it is not clearly explained why N21 is an improvement over other fragments of Hpa1 protein that has been already used and published. As the authors mention contrasting activities in N and C-terminal coiled-coil domains, it would have desirable to compare this. Or alternatively, if the main aim of work is to ensure that only the 21 residues are enough to elicit most of the defense responses, then it should be more clearly stated in the MS. Authors state that in previous work they have already shown that “two heptads of the α-helix in the N-terminal of Hpa1Xoo formed a coiled-coil domain that was responsible for HR induction in tobacco, but three heptads in the N-terminal, named N21, formed a much more stable CC structure and induced HR”. And according to what they state, the present work is done “To further reveal the functional mechanism of N21Hpa1, we performed a series of studies”. I would strongly recommend that the authors explain more clearly what are the actual objectives and hypotheses of this new work.

Besides, authors explain (although text about it requires some improvement) how transgenic tobacco plants expressing the N21 peptide were obtained, but they present results of not only these transgenic plants but also of the plants expressing the full-length protein, with no clear reference on how and when these plants were obtained. They also use E.coli, expressed recombinant proteins, but also the methods or references on how and when were obtained is missing in the text.

Other flaws or weakness that should be improved is the explanation of the defense genes induced. In the present work, it is unclear why some of the genes studied show higher induction with the N21peptide than with the full-length Hpa1 harpin, and others go in the other way around. Authors just state thattrans-N21 and trans-Hpa1 have different functions in defence responses. Our previous work demonstrated an opposite function of the coiled-coil CC domain in the N-terminal and C-terminal of HpalXoo, which may explain the differential expression of above defence response genes. This difference will be examined in future studies. I do believe that if the defense gene expression is shown in this manuscript, authors should be able to give an explanation of the contrasting behavior of the several defense genes shown here.

Moreover, according to the drought tolerance effects, I would find it interesting to present some more evidence besides the visual effects of the PEG treatment. Authors could easily measure the relative leaf water content of the leaves, and of the soil. Is there and effect on stomata opening, but if there is, why there is no effect on plant growth.  As the induction of drought tolerance is an interesting aspect of the N21 and harpin protein, I would find it interesting that the drought tolerance will be analyzed to more than the mere visual wilting of leaves, shown here. Perhaps the expression of some drought-related genes, relative water content, or biomass effects could be measured.

A minor point, the quality of the photographs shown in Figure 3 should be improved since some of the differences in disease symptoms mentioned in the text are difficult to appreciate the Figure.

Last, the English edition should be thoroughly polished in the whole manuscript.

Reviewer 3 Report

The manuscript reports the pleiotropic effects of N21 fragment of the harpin protein Hpa1 in promoting plant growth , disease and drought resistance via hypersensitive response (HR) elicitation. These biological activities were previously reported for Hpa1. The novelty resides in the association these biological activities observed in a range of plants species with the N-terminal N21 coiled-coil region of Hpa1.

While these results are of interest, they need to be backed up by more solid experimental data. For example, the authors should quantify the accumulation of pathogens in planta in a suitable number of replicates. It should be relatively straightforward to assess in planta accumulation of TMV genomic RNA in transgenic and controls plants at different times after infection. Similarly bacterial (P. carotovora carotovora) and fungal (M. fructicola) growth in planta should be assessed in transgenic and control plants as for N21-treated plants. The manuscript would be significantly improved once these experimental data are reported and a significant reduction in pathogen growth in planta observed in response to N21 expression or treatment.

Once these comments are adequately addressed, I would recommend the manuscript for publication.